**A data set of distributed global water withdrawal from 1960 to 2017**
Denghua Yan[1,2], Baisha Weng[1,2], Tianling Qin[1,2], Hao Wang[1,2], Xiangnan Li[1,2*], Yuheng Yang[1,2*], Kun Wang[1,2],
Zhenyu Lv[1,2], Jianwei Wang[1,2], Meng Li[1,2], Shan He[2], Fang Liu[2], Shanshan Liu[2], Wuxia Bi[2], Ting Xu[2], Xiaoqing
Shi[2], Zihao Man[2], Congwu Sun[2], Meiyu Liu[2], Mengke Wang[2], Yinghou Huang[2], Haoyu Long[2], Yongzhen Niu[2],
Batsuren Dorjsuren[2], Mohammed Gedefaw[2], Abel Girma[2], Asaminew Abiyu[2]
*[1] State Key Laboratory of Simulation and Regulation of Water Cycle in River Basin, China Institute of Water*
*Resources and Hydropower Research, Beijing 100038, China.*
*[2] Water Resources Department, China Institute of Water Resources and Hydropower Research, Beijing*
*100038, China.*
* Corresponding author: Xiangnan Li (lixn0555@163.com); Yuheng Yang (sduyyh@126.com).
**Abstract**
More and more high-resolution data sets are simulated worldwide and used in various research. However,
in addition to the improvement in accuracy, the practical significance of the spatial distribution of data must also
be considered. Considering that the most accurate water withdrawal data are mainly provided by the state, water
is mainly concentrated on artificial surface and cultivated land. Whenever possible, using data published by
regional or national governments and interpolating and extending them to specific land uses will maximize data
accuracy. Based on this, we provide a set of water withdrawal intensity products from 1960 to 2017 distributed
to the administrative units or the corresponding regions. The data set fills the gaps in the multi-year data set of
the accurate intensity of water withdrawal. The datasets described in this article are publicly and freely available
through the FigShare. The DOI for the data is https://doi.org/10.6084/m9.figshare.10012559.v1 (Yan et al., 2019).

**Introduction**
The requirement and utilization of water resources vary with time and geographical changes (Snow, 2005).
According to the United Nations World Water Development Report 2018, with the rapid growth of the global
population, the demand for water is expected to increase by nearly one third by 2050 (Houngbo, 2018). The
increase in water demand makes it difficult to make decisions about water allocation, and has affected the



development of industries that are critical to sustainable development, especially food and energy production.
The demand competition of water withdrawal further exacerbates the risk of regional conflicts. According to the
United Nations World Water Development Report 2018, about 1.5 billion people in 80 countries and regions,
which account for 40% of the world's total population, are under-resourced, and about 300 million people in 26
countries are extremely short of water (Peter, 2018). Studying the changes in world water withdrawal patterns,
accurately analyzing the water withdrawal intensity structure, can reveal the regional characteristics of the water
withdrawal and the regularity of water space distribution, and has important practical significance in promoting
the harmonious and sustainable development of economy and resources of the world.
Water withdrawal intensity is the main forms of water consumption and the main indicator of regional
development differences (Nouri, 2015). Water withdrawal intensity refers to the water consumption in the global,
continental, national and even smaller administrative regions, economic zones and other land areas within a
specific geographical area. The traditional water withdrawal evaluation usually takes the country as the minimum
administrative unit with low spatial resolution, and the data usually reflects the summary or average value within
a statistical unit, failing to reveal the spatial distribution differences of socio-economic data within the statistical
unit (Balk et al., 2006; Doxsey et al., 2015). With the continuous development of high-resolution grid data,
higher requirements are put forward for global grid data, and subnational data also have more application
potential. Most of the sources of water withdrawal data released by the world bank, FAO and other international
organizations are based on national governments or statistical departments. However, these data may deviate
from the real water withdrawal data of a country or region due to statistical caliber, and there is almost no
continuous long series of water withdrawal data globally. Moreover, most of the data released by international
organizations is based on national scale and lacks finer regional data. Therefore, appropriate methods are needed
to modify the data and distribute them to the corresponding spatial location.
This data set includes a set of water withdrawal intensity products distributed to the administrative units, a
set of water withdrawal intensity products distributed to artificial surface and cultivated land, and an EXCEL
file for the total amount of water withdrawal of different countries.
Data cover almost all regions of the world. Water withdrawal data include 214 national units and 616
national or sub-national units. Because of the difficulties in obtaining regional data in some countries, sub-



national data are replaced by national data. However, there may be some errors in the statistics and collection of
the original water withdrawal data, which may lead to deviations between the data set and the real water
withdrawal data. With the reference of the official data available, the accuracy of the data set is sufficient to meet
the current research. This water withdrawal data set fills the blank of complete water withdrawal sequence data
and provides products that can reflect the spatial and temporal changes of water withdrawal in the world.
Therefore, it can be used as the basic information for the study of global climate change, environmental resources,
regional economy and political decision-making. With the improvement of the accuracy of the original data
acquisition in the future, the data set can be amended and supplemented.

| Name | Acquisition Year | Source | Spatial Resolution | Format/ Pixel Type & Depth | Spatial Reference | Spatial Coverage |
|---|---|---|---|---|---|---|
| Globeland 30 | 2000/2010 | National Geomatics Center of China (NGCC) | 1" (~30m) | @ | GCS WGS 1984 | Global |
| National boundaries | 2015 | National Geomatics Center of China (NGCC) | 1km | ESRI polygon shapefile | GCS WGS 1984 | Global |
| Subnational boundaries | 2015 | National Geomatics Center of China (NGCC) | 1km | ESRI polygon shapefile | GCS WGS 1984 | Global |
| Population/Water Withdrawal | 1960-2017 | World Bank | @ | Excel | @ | Global |
| Water Withdrawal | 1960-2017 | Food and Agricultural Organization | @ | Excel | @ | Global |
| Water Withdrawal | 1960-2017 | UN data | @ | Excel | @ | Global |
| Population/ Water Withdrawal | @ | Government or statistical bureau | @ | Excel | @ | Nation |

Table 1. Input datasets, used to produce the global water withdrawal products

| Name | Temporal extent | Spatial Resolution | Format/ pixel Type & Depth | Spatial Coverage |
|---|---|---|---|---|
| *_widyear_1km | 1960~2017 | 1000 m | TIFF/flt32 | Continent |
| *_winyear_sr | 1960~2017 | 1000 m | TIFF/flt32 | Continent |
| *_Coutry_W_Data | 1960~2017 | @ | Excel | Continent |
| *_lu10 | 2000/2010 | 1000 m | TIFF/flt32 | Continent |
| *_lu80 | 2000/2010 | 1000m | TIFF/flt32 | Continent |

Table 2. The global water withdrawal products

**Methods**
In this chapter, we describe in detail the method of dataset generation, including data collection, data
modification and interpolation extension, and grid data generation.
First, the collection of water withdrawal data. Collect as much as possible of the national and sub-national
permanent population and water withdrawal data released by governments and institutions on a global scale, and
then obtain the per capita water withdrawal data.
Second, establish a national defect data interpolation model. Based on the shape of the sample data scatter
plot, determine the most appropriate curve model. Simulation modeling is implemented using EXCEL or
MATLAB.
Third, establish a sub-national defect data interpolation model. After completing the national data
interpolation, determine the best interpolated or extended model for the simulation of defect sub-national data.
Simulation modeling is implemented using EXCEL.
Fourth, create spatial distribution grids. Spread the water withdrawal intensity to the administrative unit,
and, artificial surface and cultivated land.
Fifth, data verification. We divide the measured data into calibration and verification period, re-interpolate
the data using the data of calibration period, and then verify the simulation accuracy by using the data of
verification period and the simulation.



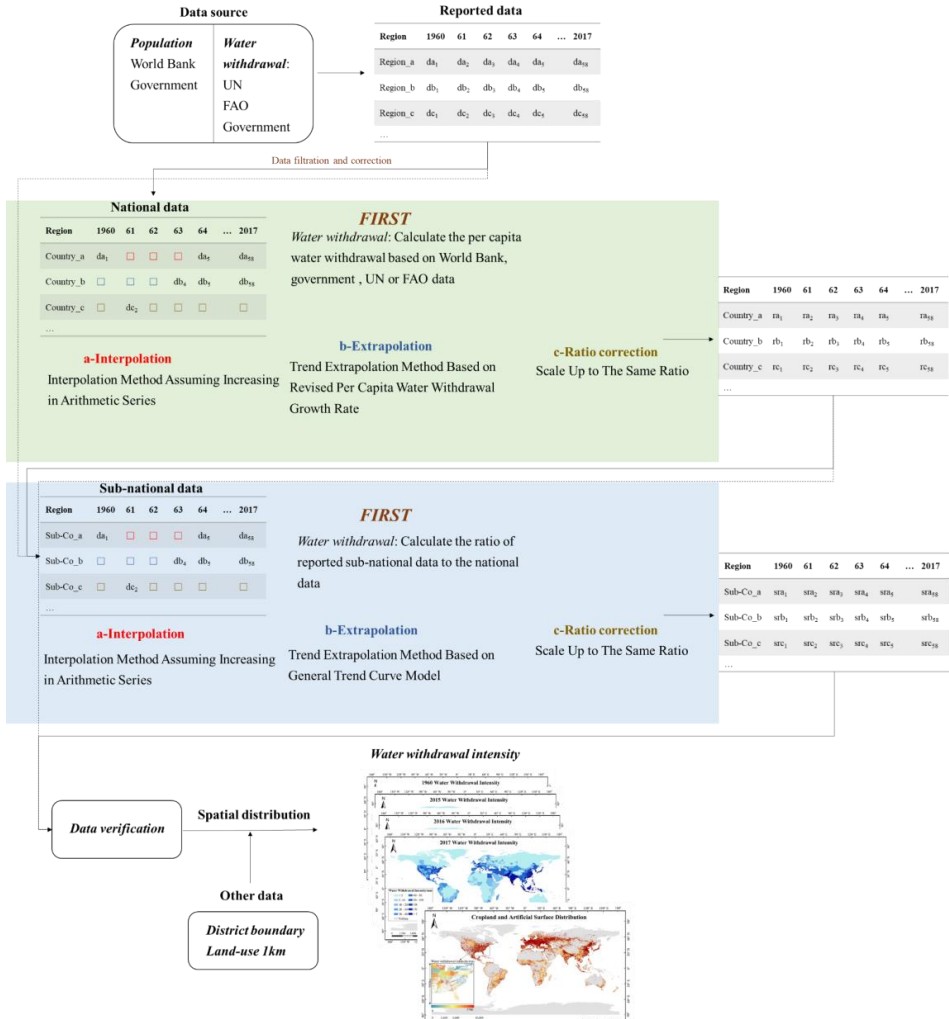


Figure 1. Schematic outline to produce the global water withdrawal products
*Data collection and pretreatment*
The data sources include government population data for xx nation and xx sub-nation, government water
withdrawal data for xx nations and xx sub-nations, national population and water withdrawal data from the World
Bank, and national population and water withdrawal data from FAO, water withdrawal data from the United
Nation, national population and water withdrawal data from Eurostat, and Globeland30 data for 2000 and 2010.
Globally, it is believed that the accuracy rate of census results obtained by counting the population of
various administrative units in the country is the highest at present when a large amount of manpower and





material resources are spent by the country itself (Honjo et al., 2015). In addition to the census conducted every
certain year, the statistical department gets high accuracy rate by calculating the overall figures according to the
sample survey of population changes and the random sample survey of fertility rate in some areas and some
units. To sum up, we believe that the data released by the government on the statistical official website is the
most reliable.

Secondly, when national population data are missing, it is generally believed that the data of the World

Bank and FAO are authoritative. When the data of World Bank and FAO are complete, the World Bank data
prevails as reference population data. When the length of World Bank data is shorter that of than FAO, the FAO
data is used as reference population data (Kumar et al., 2019).

For water withdrawal data, FAO and UN data are generally considered authoritative when government

water withdrawal data is missing. When the FAO and UN data are both complete, the FAO data is used as a
reference for water withdrawal data.
*Interpolation and extrapolation of national water withdrawal data*

The total amount of water withdrawal in various countries varies greatly, but the per capita water withdrawal

of the country generally remains within a certain range. Therefore, we first calculate the per capita water
withdrawal of the reference data, and then interpolate and extrapolate the missing per capita water withdrawal
data. The methods of can be summarized as the following five categories (Deichmann et al., 2001; Balk et al.,
2006; Tobler et al., 1997).
(1) Interpolation Method Assuming Increasing in Arithmetic Series

If discontinuities exist in per capita water withdrawal data, and the number of data increases in arithmetic

series according to the judgement, then linear interpolation method can be used based on linear model of
arithmetic series growth. This method is suitable for interval data interpolation with short interruption time and
relatively uniform data growth scale. The interpolation model is as follows:
$$P_{Nk} = [\frac{I(j)-I(i)}{j-i} \cdot (k-i) + I(i)] \cdot P_{Wk} \tag{1}$$

Where, $P_{NK}$ is the per capita water withdrawal data for the $k$ year, $i \leq k \leq j$, $I(j)$ and $I(i)$ are the ratios of government
data to reference data for the $j$ year and $i$ year, respectively.





(2) Trend Extrapolation Method Based on Revised Per Capita Water Withdrawal Growth Rate

If there are continuous points in the data, we assume that the per capita water withdrawal versus time curve

is consistent with the S curve, that is, the per capita water withdrawal shows only a slow change in the first years
and the last years. We first calculate the growth rate of per capita water withdrawal in the last two years or the
first two years, adjust the final growth rate proportionally to reflect the subsequent changes, and adjust the first
growth rate proportionally to reflect the previous changes. Equation (2) represents a method of extrapolating the
previous missing value data, and Equation (3) represents a method of extrapolating the subsequent missing value
data.
$$\begin{cases} s_i = \frac{w_i - w_{i+1}}{w_{i+1}} \\ s_{i-1} = s_i \cdot (1 - \theta) \\ w_{i-1} = w_i \cdot (1 + s_{i-1}) \end{cases} \tag{2}$$
$$\begin{cases} s_j = \frac{w_j - w_{j-1}}{w_{j-1}} \\ s_{j+1} = s_j \cdot (1 - \theta) \\ w_{j+1} = w_j \cdot (1 + s_{j+1}) \end{cases} \tag{3}$$
Where $w_{i-1}$ is the missing per capita water withdrawal value for time step $i$-1, $s_{i-1}$ is the missing reverse order
growth rate value for time step $i$-1, $w_i$ and $w_{i+1}$ are the first two known per capita water withdrawal value for
time step $i$ and $i$+1, and $s_{i-1}$ is the known reverse order growth rate value for time step $i$-1. For equation (3), $w_{j+1}$
is the missing per capita water withdrawal value for time step $j$+1, $s_{j+1}$ is the missing growth rate value for time
step $j$+1, $w_{j-1}$ and $w_j$ are the last two known per capita water withdrawal value for time step $j$ and $j$-1, and $s_j$ is the
known growth rate value for time step $j$. In order to ensure that the per capita water withdrawal in the front of
the sequence or in the latter part of the sequence does not change too fast, the equation introduces $\theta$ to represent
the correction coefficient for the growth rate, which is generally in the range of 0.1 to 0.2.
(3) Scale Up to The Same Ratio

If there is only one data collected, the per capita water withdrawal of that year will be used for all years.

(4) MATLAB Smoothing Spline Fitting

For water withdrawal data with long time spans and more data but many intervals, we use MATLAB

smoothing spline to provide smooth interpolation over time, taking into account the equilibrium of per capita
water withdrawal fluctuations.
(5) Proximity of Adjacent Region

If no national water withdrawal data is collected, based on the country's level of development and

geographic location, the per capita water withdrawal of adjacent countries with similar development levels is
selected as an approximate value for the country's per capita water withdrawal value.
*Interpolation and extrapolation of sub-national water withdrawal data*

First, the ratio of the sub-national data to the national data of the known year is calculated, and then the

interpolation and extrapolation methods are used to calculate the ratio of the missing values, and finally sub-
national data is obtained by the national data and the ratio. The methods can be summarized as the following
four categories.
(1) Interpolation Method Assuming Increasing in Arithmetic Series

If there is a discontinuity in the sub-national data, we believe that the change in the ratio of the sub-national

to national data is basically increased by the arithmetic progression. Similar to the country, then linear
interpolation method can be used based on linear model of arithmetic series growth. The interpolation model is
as follows:
$$I(j) = \frac{P_{Rj}}{P_{Nj}}, I(k) = \frac{I(j)-I(i)}{j-i} \cdot (k-i) + I(i)$$    (4)
Where, $P_{Rj}$ and $P_{Nj}$ are sub-national data and national data for j year, respectively, $I(j)$ and $I(i)$ are the known
ratios of sub-national data to national data for the *j* year and *i* year, and $I(k)$ is the ratio of the sub-nationa data to
the national data of the interpolated year.
(2) Trend Extrapolation Method Based on General Trend Curve Model

If there are continuous points in sub-national data, then the interpolation can be obtained by building a

general trend curve model based on the ratio of sub-national data to national data. General trend curve models
used in interpolation are usually conic, cubic and exponential curves, and sometimes other trend line models are
also used. Which trend line model is more suitable can be judged and selected through multiple simulation results.
For the poor fitting effect, the extrapolated year directly selects the adjacent $I(i)$ value with the known data year.
$$P_{Gk} = F(k) \cdot P_{Rk}$$    (5)
Where, $P_{Gk}$ is the sub-national data for k year, and $F(k)$ is the curve model for the ratio of sub-national data to



national data in the $k$ year.
(3) Scale Up to The Same Ratio
If there is only one year of sub-national data, then the sub-national data will be scaled up to the same ratio
according to the ratio of sub-national data to the national data of the corresponding year.
$$I = \frac{P_G}{P_R}, \quad P_{Gi} = I \cdot P_{Ri} \tag{6}$$
Where, $I$ is the ratio of sub-national data to national data.
(4) Based Entirely on Government Data
If there is a complete government data, the government data is used as the final sub-national result
*Spatial distribution*
Using remote sensing data, including Landsat 5 Thematic Mapper, Landsat 7 Enhanced Thematic Mapper
Plus (ETM+), image data of HJ-1 and local BJ-1, the Chinese Government carried out a comprehensive method
based on pixel classification, object extraction and knowledge checking in 2000 and 2010. Ten first-level remote
sensing mappings of land surface waters, wetlands, woodlands, grasslands, shrubs, artificial land surface, arable
land, glaciers and permanent snow, tundra, bare land, etc. were made in the two base years
([www.globeland30.com](www.globeland30.com)). Among them, artificial surface and cultivated land are the place where the water
withdrawal is mainly concentrated. Therefore, this paper fully considers the indicative role of specific land use
types. In the spatial distribution, it is assumed that the water is only used on artificial surface and cultivated land.
Spatial distribution, which means that the data is distributed to a meaningful area. First, based on ArgGIS
Desktop 10.2, convert the global 853 scenes land use grid into a vector format, and then extract the global
artificial surface and cultivated land. The water withdrawal intensity on the grid are expressed as follows
(Salvatore et al., 2005):
$$WU_R = \frac{U_R}{A_R}, \quad WU_{RJ} = \frac{U_R}{A_{RG}} \tag{7}$$
Where, $WU_R$ is the water withdrawal intensity of an administrative unit, respectively, $WU_{RJ}$ is the water
withdrawal intensity on the artificial surface and cultivated land of an administrative unit, $U_R$ is the water
withdrawal of an administrative unit, and $A_R$ and $A_{RG}$ are the area of an administrative unit, and the area of
artificial surface and cultivated land of an administrative unit.




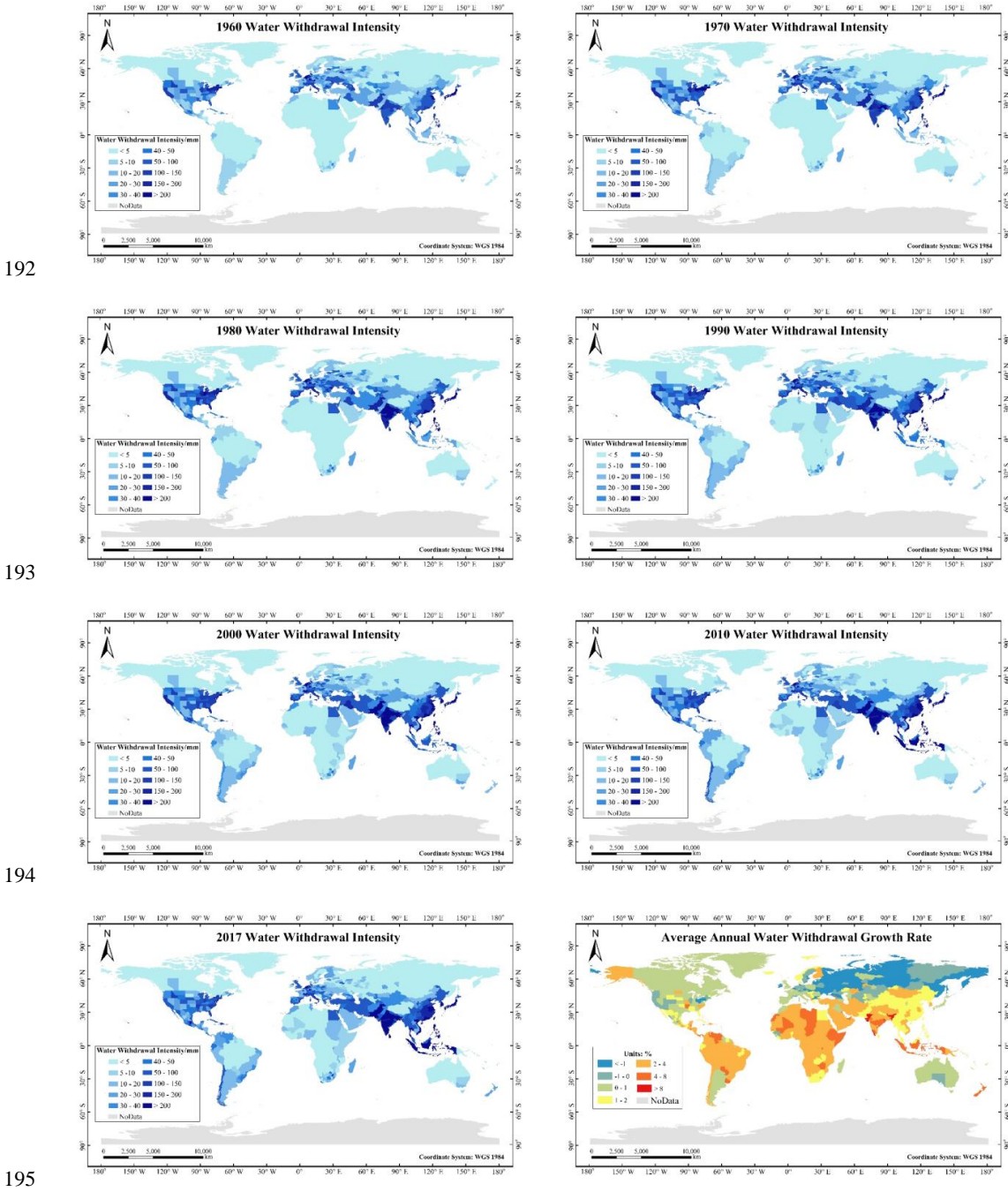

**Figure 2.** Maps of gridded regional water withdrawal intensity of administrative units for seven selected years


and the average annual water withdrawal growth rate over the study period of 1960–2017. Derived from a




combination of sub-national and national data.
**Figure 3.** Maps of gridded regional water withdrawal intensity of the artificial surface and cultivated land for



seven selected years and the average annual water withdrawal intensity growth rate over the study period of
1960–2017. Derived from a combination of sub-national and national data.
**Technical Validation**
To make the dataset more transparent, we have compiled detailed data sources for water withdrawal for
each country, and the data is available as a separate Excel spreadsheet. These data sources, including the World
Bank, FAO, the United Nations, and officially released data, are relatively accurate. Since some of the dataset
was obtained by the interpolation and extrapolation, we performed verification for the reliability of the dataset.
For the water withdrawal data, we selected several countries with more official data, and verified the
interpolation and extrapolation method, that is, selected some measured data for interpolation and extrapolation,
and compared with other measured data. China and India have a large proportion of water withdrawals in the
world and more official data, so we verified these two countries. The results show that the deviation of the data
is mostly within ±10%, and the reason for the large deviation of some points is due to the large annual fluctuation
of official data. Therefore, it can be considered that the dataset derived from the existing water withdrawal data
is accurate.

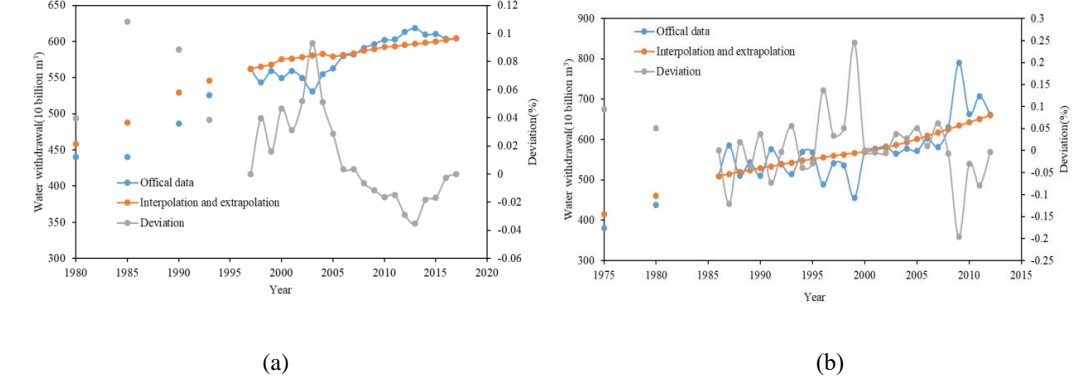


(a) (b)
Figure 4. Verification of the population data, (a) China, (b) India.
**Data availability**
The output datasets described in this article are publicly and freely available through the figshare website
(https://figshare.com/s/c360fd4227f5cb5f4873) (Yan et al., 2019). The dataset includes 4 sets of raster data and
1 sets of Excel spreadsheet data, which are published separately on each continent. Abbreviations for each



continent are as follows: NA-North America, SA-South America, EU-Europe, AS-Asia, AF-Africa, OC-Oceania.
The data includes the following:
*_winyear_sr: water withdrawal intensity grid datasets distributed on the national or sub-national
administrative units, and the unit is mm.
*_widyear_1k: water withdrawal intensity grid datasets distributed on the artificial surface and cultivated
land grids, and the unit is mm.
*_lu10: the cultivated land grid datasets and the resolution is 1km.
*_lu80: the artificial surface grid datasets and the resolution is 1km.
*_Coutry_W_Data.xlsx: original documents of national water withdrawal data, including year, World Bank
data, FAO data, UN data, government published data, and revised water withdrawal data, with a unit of 10,000m$^3$.
* indicates AF, AS, EU, NA, OC and SA.

**236   Conclusions**

The product set is designed to fill the blanks of complete water withdrawal sequence data, enhance the
accuracy and spatial variability of water withdrawal data, and can reflect the space-apace changes of water
withdrawal. The data products reveal the changes in the pattern of water withdrawal. In particular, it is of great
significance to master the scale of water withdrawal in regions where data are difficult to access to.
The product offers the water withdrawal products of the minimum administrative units and the artificial
surface-cultivated land products, and a set of EXCEL files of the revised total water consumption in different
countries. The data can be edited to meet the needs of various users.
The trend interpolation and extrapolation in the product are conducted under the assumption that the water
withdrawal maintains a certain natural growth and they cannot timely reflect the sudden changes in water
withdrawal caused by major disasters (i.e. extreme floods and earthquakes), wars and large-scale migration.
Given the mobility and flexibility of human activities, there may be some errors in the above data in many
countries/regions. Therefore, users are encouraged to make up for the error by using recently updated data or
data from specific sources.
At present, we have not collected related data of a few countries/regions, such as Mauritania, Madeira Island,
St. Helena, Christmas Island, British Indian Ocean territory, the Vatican, Svalbard Island and Jan Mayen Island,



Guadeloupe (France), St Pierre et Miquelon, Na Varsa Island, Anguilla, Montserrat, Martinique, Clipperton
Island (France), Midway Islands, Virgin Islands (British), Netherlands Antilles, United States Miscellaneous
Islands, Pitcairn Islands, Norfolk Island, Heard-und McDonald- Island, Bouvet Island, South Georgia and South
Sandwich Island, Cocos (Keeling) Islands, Prince Edward Island, Wake Island (America), French Territory in
The South, Falkland Islands, etc. Most of these areas are uninhabited or sparsely populated, so there are few
records of water withdrawal data. In the data set, they are treated as no-value areas. We intend to add more data
sets to the product in the future so as to further improve its spatial and temporal coverage.
**Acknowledgements**
This work was supported by the National Key Research and Development Program of China (No.
2016YFA0601503), National Natural Science Foundation of China (No. 51725905, No. 91547209, No.
41571037 and No. 51879276).
**Author contributions**
D.Y., B.W., T.Q. and H.W. designed the study and provided guidance. X.L. and Y.Y. drafted the manuscript.
X.L., Y.Y., K.W., Z.L., J.W. and M.L. undertook data processing and assembly. S.H., F.L., S.L., W.B., T.X., X.S.,
Z.M., C.S., M.L., M.W., Y.H., H.L., Y.N., B.D., M.G., A.G., and A.A. were involved in data collection.
**Competing interests**
The authors declare no competing interests.

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
