# Peer review of "A data set of distributed global water withdrawal from 1960 to 2017"

_Earth System Science Data, 2019_

## Referee Comment (RC1) · Anonymous Referee #1 · 4 Feb 2020

This is a thoughtful and novel paper that provides a set of water withdrawal intensity product from 1960 to 2017 at administrative level. The description is clear, the methods are innovative, and the validation is convincing. I believe that the task of data collection for this work is tremendously huge since they processed in countries around the world. This paper provides currently the first detailed water withdrawal data set and also fills in the gaps of previous research. It is of great significance for water resources utilization and watershed management in both global and regional scales. Overall the paper is relatively clearly written and I enjoyed reading it. The manuscrip is recommended for publication after the following aspects might be modified.

1. Line 17-18, "or the corresponding regions", What does this mean? Does it refer to the corresponding land use?

[Figure]

2. Line 28-29, "According to the...report". This sentence is the same as above, it can be abbreviated or omitted.

3. Line 60-61, delete the last sentence.

4. Figure 1, what does the brown FIRST mean?

5. Line 88-89, the accuracy of government data is explained here, please add some references.

6. Line 103-104, it is mentioned in the method that the per capita water withdrawal needs to be calculated, but the population data required for calculation cannot be found in the database. Please provide specific population data.

7. Line 180-182, "Among them, artificial surface and cultivated land are the place where the water withdrawal is mainly concentrated." This sentence is repeated with the following sentence.

8. Line 188, delete "respectively".

9. Figure 4, What is the unit of the right ordinate?

10. Line 241, the paper mentioned "the minimum administrative units" many times, how do you define it? To avoid different understandings, please explain it clearly.

11. The last paragraph of the Conclusions is a supplementary introduction to the data set. It might be better in the previous paragraph.

---

## Referee Comment (RC2) · Anonymous Referee #2 · 9 Feb 2020

I carefully read the paper by Yan et al.. This paper appeared to be particularly clear, well written and easy to follow. Scope and objectives are stated clearly, the description of method is systematically, and the presentation of results is rather straightforward. They describe the method of obtaining the water withdrawal with high resolution in the world from 1960 to 2017, also provide the first global water withdrawal data set that would hopefully be applied in future. I appreciate this kind of study on the relationships between global water withdrawal and land use. The authors highlight the following points from their results: 1) The need and importance of water withdrawal in the world; 2) The product set enhance the accuracy and spatial variability of water withdrawal data; 3) The product set can reflect the space-apace changes of water withdrawal. Overall, I consider that, after revisions, this paper has the potential to become a timely

and welcome addition to the literature. Specific comments: Line 16-17: I think it is necessary to indicate the resolution of the dataset in abstract. Line 23: Please refine the definition of water withdrawal and unify the use of 'water withdrawal' in the manuscript. Line 30: Change the word "account" to "accounts". Line 38-41: Please rewrite this sentence to clarify the disadvantages of traditional methods. Line 45-46: In abstract: "... regional or national governments and interpolating and extending them to specific land uses will maximize data accuracy", this sentence is the premise of the manuscript, should explain it in detail in the introduction. Line 47-48: There are some speculations in these sentences. Please add relevant references. Line 71-72: "...using EXCEL or MATLAB", please describe briefly the difference between the two software. Also, the version of the software should be mentioned. Line 87: The source of Globeland30 should be mentioned in the manuscript. Line 95-101: Also, there are some speculations in these sentences. They should be supported by reference. Line 102: This part is the key to the manuscript. I recommend to add it to Figure 1, which could facilitate further understanding of the choice of data interpolation method. Line 174: If I get through it well, the artificial surface part represents industrial and domestic water, and cultivated land represents most agricultural water, while this part still needs to add references to support your manuscript. Line 192-205: The comparison is not intuitive between the two pictures. I recommend keeping the division interval of the two figures consistent, so that the reader can intuitively learn the advantages of the new spreading method. Line 204-212: In this part, I can't distinguish whether the author only verified these two countries or use these two countries as examples for research. Line 220: Also, I'm not sure if the units in Figure 4 are correct. The manuscript mentions that the fitted water withdrawal differs from the official data by less than 10%, but it looks less than 0.25% in the figure.

---

## Referee Comment (RC3) · Anonymous Referee #3 · 27 Feb 2020

Review ESSD-2019-224, global water withdrawal

Authors address an interesting global water issue with a potentially useful data product but the products themselves and the presentation in the manuscript fail - by a very long way - to meet ESSD standards.

The Figshare link allows users easy access but the presented data remain almost useless. THE FILES CONTAIN NO METADATA!! A user finds no column headings, no introductory lines (rows) of explanation or attribution, nothing to help the user know what he or she looks at. More information comes from the filename (e.g. Data_W_Sources) than from the files themselves. Most folders include only maps, not referenced or sorted except by date. Data availability section in the manuscript (lines 221 to 235) provides only minimal guidance. A large set of guidelines and taxonomies exist for surface water hydrology - the authors have neither used nor referenced any of that. Authors should look at almost any other ESSD product to find very good examples of how to present and record metadata. Not acceptable in present form.

Entire text very weak, disorganized, confusing. This review highlights dozens of issues below. Basically, authors show some skill in mapping and interpolation, but have not shown understanding or skill with data or description to help users. Does not meet standards expected for a top data journal.

Line 29 - "United Nations World Water Development Report 2018": after authors have cited this report once (line 24) they should use a standard acronym. The report itself suggests a proper citation, which these authors should use.

Line 33 - "regularity of water space distribution" - What does this mean? Regularity in time or space? Regularity in classification? This entire introductory paragraph basically repeats the same issues two or three times. Whatever motivation might exist remains hidden by or obscured by random language.

Line 35 - "withdrawal intensity is the main forms of water consumption" - Singular / plural confusion, happens frequently throughout the manuscript, often confuses the reader about what authors intend.

Lines 35 to 48: this entire paragraph reminds user that authors have produced a geographic data product, not a hydrologic data product. But, from the title, authors implied a hydrological product? Authors should make clear their intent, their tools and their skills. This is a GIS / mapping exercise, not an effort to produce a valid global hydrologic data product.

Line 54 - "errors in the statistics and collection of the original water withdrawal data": most users will know that accurate data on water supply or use remains highly restricted and highly distorted by most countries. Here the hydrology / geography confusion arises clearly: what use does a higher spatial resolution product have if the underlying data remain almost completely unreliable. These authors with their spatial

GIS skills never address the fundamental issue of the quality or availability of the basic hydrological data?

Line 56 - "With the reference of the official data available, the accuracy of the data set is sufficient to meet the current research": yes, the authors can assemble and provide nice maps but the fundamental data remain almost useless.

Line 58 - "products that can reflect the spatial and temporal changes of water withdrawal in the world": No, strongly disagree. New maps from bad data, not an overall improvement.

Line 60 - "improvement of the accuracy of the original data": improvement of spatial resolution does not equal improvement of accuracy.

Line 62, Table 1: The FAO data products are notoriously unreliable. Likewise for other UN sources. Chinese sources seem perhaps interesting and useful but authors provide no publications, validations, or reliability analyses?

Lines 64 to 205, Methods: This is basically a GIS, data-filling, data interpolation exercise, to fill in missing national reports and then to fill in spatial gaps. The authors, who may have skill in using these products and tools, give no indication that they understand, much less question, the underlying data. I repeat: better maps of unreliable data do not result in better data.

Lines 175 to 182: Here the authors provide a weak description of GlobeLand30, the Chinese remote sensing product (based on USGS Landsat). No details, no references, no uncertainties, nothing accessible or open access that can convince other users of the quality of this product.

Line 182 - "water is only used on artificial surface and cultivated land": Reader / user never learns what the authors mean by the term 'artificial surface' (reservoir?, impervious urban pavements?, compacted drought-impacted land with high run-off?) but the assumption stated in this phrase is almost certainly false. The authors demonstrate no understanding whatsoever of surface hydrology!

Line 206 and following - Technical validation: Validation consists of comparing their interpolated products to two countries, India and China, for which they have data of higher spatial resolution. But from exactly the same FAO/UN national report source data! How does that represent independent data? Many other data products, including those derived from satellites (e.g LandSat or others), exist, for which these authors could at least construct some intercomparison maps. Not one quantitive assessment (run-off, storage, retention, etc.) in useful hydrological terms anywhere in the entire manuscript. Validation, uncertainty, reliability completely absent.

Line 216 - "dataset derived from the existing water withdrawal data is accurate" Authors may believe this, but due to complete ignorance of surface hydrology, reader /

user can only accept that maps might prove faithful to national reports but that underlying data has neither validity nor accuracy. Authors can neither make nor verify any statement about accuracy. Large community of hydrologists sharing data through ESSD will not find this product in any way useful.

Line 237 - "fill the blanks of complete water withdrawal sequence data, enhance the accuracy and spatial variability of water withdrawal data, and can reflect the space-apace changes of water withdrawal." Complete = no. Accuracy = no. Spatial variability = a product only of their interpolation tools, never verified. Space-apace = ????

Line 270 and following: 13 references, 4 of which represent unreviewed UN or private technical reports. If well and carefully done, a product like this could have reference to dozens of primary and validation sources and to literally hundreds of hydrological and social applications. The authors seem aware of none of that, not even the vast remote sensing or interpolation literature.

---

## Author Comment (AC1) · 27 Feb 2020

Comments and Suggestions for Authors This paper provides a set of water withdrawal intensity products from 1960 to 2017 at administrative units. This paper is clearly structured and easy follow, with a good view of their methods and a well performed validation. I believe that the task of data collection for this work is tremendously huge since they have many countries to process. This work has great significance for global and regional water resources utilization and watershed management. This is currently the first detailed water withdrawal data set and it also fills in the gaps of previous research. I think this manuscript needs some minor revisions before published:

Point 1: Line 17-18, "or the corresponding regions", What does this mean? Does this

refer to the corresponding land use? Response 1: In this study, we defined the concept of water withdrawal intensity. In the spatial distribution, it is assumed that the water withdrawal occurs only on artificial surface and cultivated land. Therefore, we have considered differences in spatial distribution within the statistical unit. Point 2: Line 28-29, "According to the...report". This sentence is the same as above, it can be abbreviated or omitted. Response 2: According to the comments, we rewrite this sentence as follows: According to the Report 2018, about 1.5 billion people in 80 countries and regions, which account for 40% of the world's total population, are under-resourced, and about 300 million people in 26 countries are extremely short of water. Point 3: Line 60-61, delete the last sentence. Response 3: According to the comments, we deleted this sentence in the Introduction. For the future improvement of the data set, we briefly introduced in the Conclusion. Point 4: Figure 1, what does the brown FIRST mean? Response 4: FIRST refers to the first step in data processing. In the article, we detailed the steps of data production. There are different strategies for processing national and subnational water withdrawal data. For national data, we need to first calculate per capita water withdrawal data, and for sub-national data, we need to first calculate the ratio of the sub-national data to the national data. Therefore, in the figure, we have emphasized the "FIRST step". Point 5: Line 88-89, the accuracy of government data is explained here, please add some references. Response 5: Based on comments, we have added relevant references. Point 6: Line 103-104, it is mentioned in the method that the per capita water withdrawal needs to be calculated, but the population data required for calculation are not seen in the database. Please provide specific population data. Response 6: For demographic data, we comprehensively collected demographic data sets released by the World Bank and the government. In subsequent modifications, we will upload these data to the same database as the water withdrawal data set. Point 7: Line180-182, "Among them, artificial surface and cultivated land are the place where the water withdrawal is mainly concentrated." This sentence is repeated with the following sentence. Response 7: Based on the comments, we deleted the subsequent repeated sentences and rewritten them as follows: Among them, artificial

surface and cultivated land are the place where the water withdrawal is mainly concentrated. Therefore, this paper fully considers the indicative role of specific land use types. Point 8: Line 188, delete "respectively". Response 8: We have modified it based on comments. Point 9: Figure 4, What is the unit of the right ordinate? Response 9: We have mistaken the ordinate unit in the first draft. The unit should be "1", not "%". Point 10: Line 241, the paper mentioned "the minimum administrative units" many times, what does this mean? Different readers may understand differently, please explain clearly. Response 10: Thank you for your comments. In this study, the minimum administrative unit refers to the data of the smallest administrative region in a country that we can collect, most of which are provincial data, and some are watershed data. Point 11: The last paragraph of the Conclusions is a supplementary introduction to the data set. It can be placed in the previous paragraph. Response 11: Thank you for your comments, we will adjust this in subsequent revisions.

---

## Author Comment (AC2) · 27 Feb 2020

Comments and Suggestions for Authors I carefully read the paper by Yan et al.. This paper appeared to be particularly clear, well written and easy to follow. Scope and objectives are stated clearly, the description of method is systematically, and the presentation of results is rather straightforward. They describe the method of obtaining the water withdrawal with high resolution in the world from 1960 to 2017, also provide the first global water withdrawal data set that would hopefully be applied in future. I appreciate this kind of study on the relationships between global water withdrawal and land use. The authors highlight the following points from their results: 1) The need and importance of water withdrawal in the world; 2) The product set enhance the accuracy and spatial variability of water withdrawal data; 3) The product set can reflect the

space-apace changes of water withdrawal. Overall, I consider that, after revisions, this paper has the potential to become a timely and welcome addition to the literature.

Point 1: Line 16-17: I think it is necessary to indicate the resolution of the dataset in abstract. Response 1: The spatial resolution of the dataset is 1km×1km. Based on the comments, we will introduce the resolution in abstract in subsequent amendments. Point 2: Line 23: Please refine the definition of water withdrawal and unify the use of 'water withdrawal' in the manuscript. Response 2: Based on the comments, we added the definition of the water withdrawal. According to the comments, we have added the definition of water withdrawal, which refers to the amount of water used by water users. Water is usually provided by the water supply unit or it can be obtained by water users directly from rivers, lakes, reservoirs (ponds) or underground Point 3: Line 30: Change the word "account" to "accounts". Response 3: We have modified it based on comments. Point 4: Line 38-41: Please rewrite this sentence to clarify the disadvantages of traditional methods. Response 4: The traditional water withdrawal evaluation usually takes the country as the minimum administrative unit and fails to reflect the exact geographic location where the water withdrawal occurred. Point 5: Line 45-46: In abstract: "... regional or national governments and interpolating and extending them to specific land uses will maximize data accuracy", this sentence is the premise of the manuscript, should explain it in detail in the introduction. Response 5: Traditional data often only considers administrative units, and this study considers not only administrative units but also actual land use. Therefore, appropriate methods are needed to modify the data and distribute them to the corresponding spatial location. Thank you for your comments, we will explain it in detail in the subsequent amendments Point 6: Line 47-48: There are some speculations in these sentences. Please add relevant references. Response 6: Based on comments, we have added relevant references. Point 7: Line 71-72: "…using EXCEL or MATLAB", please describe briefly the difference between the two software. Also, the version of the software should be mentioned. Response 7: In the study, the version of EXCEL is 2013 and the version of MATLAB is 2018. In the study, we used two kinds of software. We converted the data processing methods to

EXCEL, and uploaded it to the database. Point 8: Line 87: The source of Globeland30 should be mentioned in the manuscript. Response 8: The Globeland30 data is provided by the National Geographic Information Center. We have also uploaded specific artificial surface and cultivated land data to the database. Point 9: Line 95-101: Also, there are some speculations in these sentences. They should be supported by reference. Response 9: Based on comments, we have added relevant references. Point 10: Line 102: This part is the key to the manuscript. I recommend to add it to Figure 1, which could facilitate further understanding of the choice of data interpolation method. Response 10: In fact, these data processing methods are all reflected in Figure 1, but Figure 1 is too simple at present, and we will enrich Figure 1 according to your comments. Point 11: Line 174: If I get through it well, the artificial surface part represents industrial and domestic water, and cultivated land represents most agricultural water, while this part still needs to add references to support your manuscript. Response 11: Based on comments, we have added relevant references. Point 12: Line 192-205: The comparison is not intuitive between the two pictures. I recommend keeping the division interval of the two figures consistent, so that the reader can intuitively learn the advantages of the new spreading method. Response 12: Thank you for your comments. We will modify the division interval of the legend according to the comments. Point 13: Line 204-212: In this part, I can't distinguish whether the author only verified these two countries or use these two countries as examples for research. Response 10: In this section, we mainly verify our data processing methods. In fact, we have carried out detailed data calculations for each country. Due to the excessive number of countries, we have uploaded specific data processing files to the database. Point 14: Line 220: Also, I'm not sure if the units in Figure 4 are correct. The manuscript mentions that the fitted water withdrawal differs from the official data by less than 10%, but it looks less than 0.25% in the figure. Response 14: We have mistaken the ordinate unit in the first draft. The unit should be "1", not "%". Actually, the fitted water withdrawal differs from the official data by less than 10%.

---

## Author Comment (AC3) · 1 Apr 2020

Reply to the reviewer comments Responses to Reviewer #3: Dear Reviewer: Thank you for your comments concerning our manuscript entitled "A dataset of distributed global water withdrawal from 1960 to 2017" (ID: ESSD-2019-224). Those comments are all valuable and very helpful for revising and improving our paper, as well as the important guiding significance to our researches. We have studied comments carefully and modified the dataset which we hope meet with approval. The main corrections in the dataset and the responds to your comments are as follows:

Line-by-line comments: Referee comment: 1. The Figshare link allows users easy access but the presented data remain almost useless. THE FILES CONTAIN NO META-

[Figure]

DATA!! A user finds no column headings, no introductory lines (rows) of explanation or attribution, nothing to help the user know what he or she looks at. More information comes from the filename (e.g. Data_W_Sources) than from the files themselves. Most folders include only maps, not referenced or sorted except by date. Data availability section in the manuscript (lines 221 to 235) provides only minimal guidance. A large set of guidelines and taxonomies exist for surface water hydrology - the authors have neither used nor referenced any of that. Authors should look at almost any other ESSD product to find very good examples of how to present and record metadata. Not acceptable in present form. Entire text very weak, disorganized, confusing. This review highlights dozens of issues below. Basically, authors show some skill in mapping and interpolation, but have not shown understanding or skill with data or description to help users. Does not meet standards expected for a top data journal. Author's response: We introduced the attribute information of the dataset in the manuscript, which may not be detailed enough. Based on your comments, we will add metadata and provide more detailed data guidance in the manuscript. Author's changes to the manuscript: We will add the metadata, column headings, lines of explanation and we will provide more detailed data guidance in the data availability section.

Referee comment: 2. Line 29 - "United Nations World Water Development Report 2018": after authors have cited this report once (line 24) they should use a standard acronym. The report itself suggests a proper citation, which these authors should use. Author's response: Thank you. We will include a standard acronym and reference. Author's changes to the manuscript: In the revision, we will use a standard acronym and format of reference.

Referee comment: 3. Line 33 - "regularity of water space distribution" - What does this mean? Regularity in time or space? Regularity in classification? This entire introductory paragraph basically repeats the same issues two or three times. Whatever motivation might exist remains hidden by or obscured by random language. Author's response: What we want to express is that by studying changes in global water with-
drawal, accurately analyzing the structure of water withdrawal intensity, revealing the regional characteristics and spatial distribution of water withdrawal. Author's changes to the manuscript: That sentence will be changed to "regularity of the spatial and temporal distribution of water withdrawal".

Referee comment: 4. Line 35 - "withdrawal intensity is the main forms of water consumption" - Singular / plural confusion, happens frequently throughout the manuscript, often confuses the reader about what authors intend. Author's response: We will send the manuscript to a professional linguist for editing to ensure the readability of the manuscript. Author's changes to the manuscript: That sentence will be changed to "withdrawal intensity is the main form of water consumption".

Referee comment: 5. Lines 35 to 48: this entire paragraph reminds user that authors have produced a geographic data product, not a hydrologic data product. But, from the title, authors implied a hydrological product? Authors should make clear their intent, their tools and their skills. This is a GIS / mapping exercise, not an effort to produce a valid global hydrologic data product. Author's response: I think the hydrological data must contain the corresponding geographic information to be meaningful. There is no clear boundary between geographic data product and hydrologic data product, and all hydrologic data products need the support of geographic data. Author's changes to the manuscript: We will make clear the intent of manuscript in introduction.

Referee comment: 6. Line 54 - "errors in the statistics and collection of the original water withdrawal data": most users will know that accurate data on water supply or use remains highly restricted and highly distorted by most countries. Here the hydrology / geography confusion arises clearly: what use does a higher spatial resolution product have if the underlying data remain almost completely unreliable. These authors with their spatial GIS skills never address the fundamental issue of the quality or availability of the basic hydrological data? 7. Line 56 - "With the reference of the official data available, the accuracy of the data set is sufficient to meet the current research": yes, the authors can assemble and provide nice maps but the fundamental data remain almost

useless. 8. Line 58 - "products that can reflect the spatial and temporal changes of water withdrawal in the world": No, strongly disagree. New maps from bad data, not an overall improvement. Author's response: First, all data cannot be guaranteed to be 100% accurate, but it is considered reliable as long as the data is within the allowed error range. There is almost no continuous long series of water withdrawal data globally, which leads to underestimation or overestimation in the analysis of global water consumption. Without global water withdrawal data, biases can arise in assessing social, economic and ecological water requirement and in developing water management strategies. Moreover, water withdrawal is not just a physical concept, and it contains a large number of socio-economic implications, so it is impossible for any research to calculate the water withdrawal of the world or even a region with 100% accuracy. The significance of this study is not to provide a set of datasets with 100% accuracy, but to provide a complete and credible data product based on the fact that there is currently no global water withdrawal data set. Of course, with the gradual increase of data collection, the accuracy of this data product can continue to be improved in the future. Author's changes to the manuscript: None

Referee comment: 9. Line 60 - "improvement of the accuracy of the original data": improvement of spatial resolution does not equal improvement of accuracy. Author's response: I acknowledge that the reviewer's view that "improvement of spatial resolution does not equal improvement of accuracy" is correct. It is clear, however, that the reviewer did not carefully review the manuscript. What we said on line 60 of the manuscript is that in the future, when the government releases data with better accuracy (e.g., water withdrawal data by region), we can further improve the accuracy of the dataset. Author's changes to the manuscript: None

Referee comment: 10. Line 62, Table 1: The FAO data products are notoriously unreliable. Likewise for other UN sources. Chinese sources seem perhaps interesting and useful but authors provide no publications, validations, or reliability analyses? Author's response: Although the data of FAO and other international organizations do

have some deviations from the real data, the data is to a certain extent credible. A large amount of current water research worldwide is also based on existing FAO and other data. In addition, for more than 200 countries, we have collected a large amount of data released by national government departments. The credibility of these data is often higher than that of FAO. The introduction of these data sources and download links have been uploaded to the database of this study (lines 207 to 208). According to your opinion, we will supplement references in the manuscript with Chinese sources, validations, and reliability analyses. Author's changes to the manuscript: We will supplement references in the manuscript with Chinese sources, validations, and reliability analyses

Referee comment: 11. Lines 175 to 182: Here the authors provide a weak description of GlobeLand30, the Chinese remote sensing product (based on USGS Landsat). No details, no references, no uncertainties, nothing accessible or open access that can convince other users of the quality of this product. Author's response: The classified images used for the development of GlobeLand30-2010 data are mainly 30-meter multispectral images, including the United States Landsat TM5, ETM+ multispectral images, and the China Environmental Disaster Reduction Satellite (HJ-1). In the verification process, a total of more than 150,000 test samples in 9 categories were selected for accuracy evaluation. The overall accuracy of GlobeLand30-2010 data was 83.51%, and the Kappa coefficient was 0.78. The method of obtaining data and listing it in the manuscript, reviewers can download the data through this website. Since this set of data is not original, it was not uploaded to the database. Author's changes to the manuscript: We will add details, references and uncertainties about GlobeLand30 in the manuscript.

Referee comment: 12. Line 182 - "water is only used on artificial surface and cultivated land": Reader / user never learns what the authors mean by the term 'artificial surface' (reservoir?, impervious urban pavements?, compacted drought-impacted land with high run-off?) but the assumption stated in this phrase is almost certainly false.
The authors demonstrate no understanding whatsoever of surface hydrology! Author's response: Artificial surface refers to the surface formed by artificial construction activities, including various residential areas such as towns, industries, mines, and transportation facilities. The artificial surface mainly includes industrial and domestic water, and the cultivated land mainly includes agricultural water, so we are confident that the assumptions we have listed are correct. Author's changes to the manuscript: We will add the definition of artificial surface.

Referee comment: 13. Line 206 and following - Technical validation: Validation consists of comparing their interpolated products to two countries, India and China, for which they have data of higher spatial resolution. But from exactly the same FAO/UN national report source data! How does that represent independent data? Many other data products, including those derived from satellites (e.g LandSat or others), exist, for which these authors could at least construct some intercomparison maps. Not one quantitive assessment (run-off, storage, retention, etc.) in useful hydrological terms anywhere in the entire manuscript. Validation, uncertainty, reliability completely absent. Author's response: When we verified the data, we mainly verified our interpolation and extrapolation methods. For China and India, their data sources are from the data released by their governments, not the same data source (lines 211 to 213). We first assume that there are no data for some years, then use our method to obtain the data for these years and compare them with real data. Of course, the comparison in the manuscript is indeed insufficient, and we will add more data verification in the future. Author's changes to the manuscript: We will add more detailed data validation.

Referee comment: 14. Lines 64 to 205, Methods: This is basically a GIS, data-filling, data interpolation exercise, to fill in missing national reports and then to fill in spatial gaps. The authors, who may have skill in using these products and tools, give no indication that they understand, much less question, the underlying data. I repeat: better maps of unreliable data do not result in better data. 15. Line 216 - "dataset derived from the existing water withdrawal data is accurate" Authors may believe this,

but due to complete ignorance of surface hydrology, reader /user can only accept that maps might prove faithful to national reports but that underlying data has neither validity nor accuracy. Authors can neither make nor verify any statement about accuracy. Large community of hydrologists sharing data through ESSD will not find this product in any way useful. 16. Line 237 - "fill the blanks of complete water withdrawal sequence data, enhance the accuracy and spatial variability of water withdrawal data, and can reflect the spaceapace changes of water withdrawal." Complete = no. Accuracy = no. Spatial variability = a product only of their interpolation tools, never verified. Space-apace = ???? Author's response: As we mentioned earlier, by collecting and compiling a large amount of data in more than 200 countries, we can trust that the water withdrawal dataset based on government data / FAO data and other sources is accurate. We will add more detailed verification and instructions in subsequent revisions. Author's changes to the manuscript: We will add more detailed verification and instructions in subsequent revisions.

Referee comment: 17. Line 270 and following: 13 references, 4 of which represent unreviewed UN or private technical reports. If well and carefully done, a product like this could have reference to dozens of primary and validation sources and to literally hundreds of hydrological and social applications. The authors seem aware of none of that, not even the vast remote sensing or interpolation literature. Author's changes to the manuscript: We will add references in the manuscript.

---

## Editor Comment (EC1) · Ge Peng (Editor) · 27 Apr 2020

Reject. In the current form, the quality of the manuscript and data does not meet ESSD's standards. Constructed comments provided by the reviewers during the discussion of this manuscript should be used to help improve the quality of the manuscript and data.